# Reassessment of *Dyfrolomyces* and Four New Species of *Melomastia* from Olive (*Olea europaea*) in Sichuan Province, China

**DOI:** 10.3390/jof8010076

**Published:** 2022-01-13

**Authors:** Wen-Li Li, Sajeewa S. N. Maharachchikumbura, Ratchadawan Cheewangkoon, Jian-Kui Liu

**Affiliations:** 1Center for Informational Biology, School of Life Science and Technology, University of Electronic Science and Technology of China, Chengdu 611731, China; wendy316@tom.com (W.-L.L.); sajeewa83@yahoo.com (S.S.N.M.); 2Department of Entomology and Plant Pathology, Faculty of Agriculture, Chiang Mai University, Chiang Mai 50200, Thailand; ratchadawan.c@cmu.ac.th

**Keywords:** 15 new taxa, Dothideomycetes, morphology, multigene, phylogeny

## Abstract

*Pleurotremataceae* species are saprobes on decaying wood in terrestrial, mangrove, and freshwater habitats. The generic boundary of the family has traditionally been based on morphology. All genera of *Pleurotremataceae* have a high degree of morphological overlap, of which the generic circumscription of *Melomastia* and *Dyfrolomyces* has not been well resolved. Thus, the delimitation of genera has always been challenging. *Melomastia* traditionally differs from *Dyfrolomyces* in having 2-septate, oblong, with obtuse-ends ascospores. These main characteristics have been used to distinguish *Melomastia* from *Dyfrolomyces* for a long time. However, the above characteristics sometimes overlap among *Dyfrolomyces* and *Melomastia* species. Based on the morphology and multigene phylogeny with newly obtained data, we synonymized *Dyfrolomyces* under *Melomastia* following up-to-date results. Four novel species (i.e., *Melomastia fusispora*, *M. oleae*, *M. sichuanensis* and *M. winteri*) collected from the dead branches of *Olea europaea* L. in Chengdu Olive Base, Sichuan Province in China are introduced based on detailed morphological characterization and phylogenetic analyses of sequences based on nuclear ribosomal (LSU and SSU) and protein-coding gene (*tef1-α*). The 11 new combinations proposed are *Melomastia aquatica* (=*Dyfrolomyces aquaticus*), *M. chromolaenae* (=*D. chromolaenae*), *M. distoseptata* (=*D. distoseptatus*), *M. mangrovei* (=*D. mangrovei*), *M. marinospora* (=*D. marinosporus*), *M. neothailandica* (=*D. neothailandicus*), *M. phetchaburiensis* (=*D. phetchaburiensis*), *M. sinensis* (=*D. sinensis*), *M. thailandica* (=*D. thailandica*), *M. thamplaensis* (=*D. thamplaensis*) and *M. tiomanensis (*=*D. tiomanensis)*.

## 1. Introduction

The family *Pleurotremataceae* was introduced by Watson [1] for the monotypic genus *Pleurotrema* Müll. Arg. with *P. polysemum* as the type of species. The family is characterized by a clypeus on the substrate, immersed ascomata and multi-septate ascospores, with or without a sheath in bitunicate/fissitunicate asci [1,2]. Barr assigned *Pleurotremataceae* to *Xylariales* and accepted five genera, viz. *Daruvedia* Dennis., *Melomastia* Nitschke ex Sacc., *Phomatospora* Sacc., *Pleurotrema* Müll. Arg. and *Saccardoella* Speg., based on their unitunicate and non-fissitunicate ascal characteristics [2]. Later, *Daruvedia* and *Phomatospora* were transferred to *Pleosporales* and *Phomatosporales* based on DNA sequence data, respectively [3,4]. *Melomastia* and *Saccardoella* were referred to Ascomycota genera *incertae sedis* [5,6], while *Pleurotrema* was retained in *Pyrenulales* [7]. The familial placement of *Pleurotrema* has been controversial due to the confusion over the nature of the asci, as they are neither typically unitunicate nor bitunicate [8,9]. Thus, *Pleurotrema* has been classified in the *Xylariales* [10], *Pyrenulales* [11] and *Chaetosphaeriales* [12]. The type of species *P. polysemum* was re-examined to confirm the placement of *Pleurotrema* [2,13]. The bitunicate, non-fissitunicate and not lichenized asci of *Pleurotrema* determined that it has morphological affinities to the genera *Melomastia* and *Saccardoella*. Based on the above evidence, Maharachchikumbura et al. [13] excluded *Pleurotremataceae* from Sordariomycetes and placed it in Dothideomycetes.

Currently, three genera: *Dyfrolomyces* K. D. Hyde, *Melomastia* Nitschke ex Sacc and *Pleurotrema* are accepted in *Pleurotremataceae* [14,15]. The genus *Melomastia* was introduced by Saccardo [16] to accommodate *M. mastoidea* (=*Melomastia friesii* Nitschke) and is characterized by globose ascomata with erumpent apex, cylindrical asci and 2-septate, ovoid ascospores with or without a gelatinous sheath [2,17]. As the placement of the type species of *Melomastia* within Ascomycota remained unresolved and was based solely on morphological observations, this genus was classified under Ascomycota genera *incertae sedis* [13]. Norphanphoun et al. [18] introduced *Melomastia italica* with sequence data and assigned *Melomastia* in *Pleurotremataceae*, while *Dyfrolomyces maolanensis* was synonymized under *M.*
*maolanensis* based on phylogeny and morphological characteristics. Subsequently, two species from *Clematis* sp., *M. clematidis* and *M. fulvicoma* were introduced by Phukhamsakda et al. [19]. Currently, 44 epithets of *Melomastia* are listed in Index Fungorum (http://www.indexfungorum.org/Names/Names.asp, accessed on 1 December 2021), though most of them lack sequence data.

Pang et al. [20] established the family *Dyfrolomycetaceae* with the type of genus *Dyfrolomyces* to accommodate *D. tiomanensis* and accept three marine species transferred from *Saccardoella* (i.e., *Saccardoella mangrovei*, *S. marinospora* and *S. rhizophorae*). Subsequently, *Dyfrolomycetaceae* was raised as the order *Dyfrolomycetales* [21]. Maharachchikumbura et al. [13] synonymized *Dyfrolomycetaceae* under *Pleurotremataceae* based on morphological investigations. *Dyfrolomyces* is charactered by immersed, clypeate, globose ascomata, 8-spored, bitunicate, fissitunicate, cylindrical asci with relatively short pedicel, and overlapping uniseriate, fusiform, hyaline, multi-septate ascospore with or without a gelatinous sheath, which is very similar to *Melomastia*. However, the affiliation between these two genera is unclear due to the lack of sequence data of *Melomastia* species. With the description of *Melomastia italica*, *Melomastia* was revealed as closely related to *Dyfrolomyces*, and there was no significant support in the molecular phylogenies to differentiate *Melomastia* from *Dyfrolomyces* [18]. Presently, 11 *Dyfrolomyces* species have been accepted in the genus.

During a survey of micro-fungi on olive in Sichuan Province, China, we collected nine pleurotrema-like taxa from *Olea europaea*. Four new species, viz. *Melomastia fusispora*, *M. oleae*, *M. sichuanensis* and *M. winteri* were introduced. Moreover, we evaluated the morphology and phylogeny of accepted species in *Dyfrolomyces* and *Melomastia,* and synonymized *Dyfrolomyces* under *Melomastia.*

## 2. Materials & Methods

### 2.1. Isolation and Morphology

The dead branches of *Olea europaea* were collected from Chengdu Olive Base, Shuangliu District, Chengdu City, Sichuan Province, China, in 2020. The samples were transported to a laboratory in paper envelopes. Examination and observation of samples were carried out as detailed by Senanayake et al. [22]. Measurements were made with the Tarosoft (R) Image Framework program v. 0.9.7, following Liu et al. [23]. 

Single spore isolations were prepared following the method of Chomnunti et al. [24]. Germinating spores were transferred aseptically onto fresh potato dextrose agar (PDA) plates and incubated at room temperature in daylight. The strains isolated in this study were deposited at the China General Microbiological Culture Collection Center (CGMCC), Beijing, China and the University of Electronic Science and Technology Culture Collection (UESTCC), Chengdu, China. Herbarium specimens were deposited at Herbarium of Cryptogams, Kunming Institute of Botany Academia Sinica (KUN-HKAS), Kunming, China and Herbarium, University of Electronic Science and Technology (HUEST), Chengdu, China.

### 2.2. DNA Extraction, PCR Amplification and Sequencing

Genomic DNA was extracted from 14d-old fungal colonies growing on PDA at 25 °C using the EZ geneTM fungal gDNA kit (GD2416) according to the manufacturer’s instructions. Phylogenetic analyses were conducted using partial sequences of the large subunit rDNA (28S, LSU), small subunit rDNA (18S, SSU) and the translation elongation factor-1 alpha (*tef1-α*) using the primer pairs LR0R/LR5 [25], NS1/NS4 [26] and 983F/2218R [27], respectively. Polymerase chain reaction (PCR) amplifications were conducted in 25 µL reaction volumes. The PCR mixture consisted of 2× PCR MasterMix (12.5 µL) (Tsingke Co., China), genomic DNA (1 µL), each primer (1 µL) and ddH2O (9.5 µL). The PCR thermal cycle program for LSU, SSU and *tef1-α* amplification was performed using the following parameters: 94 °C for 3 min, followed by 35 cycles of denaturation at 94 °C for 30 s, annealing at 56 °C for 50 s, elongation at 72 °C for 1 min and a final extension at 72 °C for 10 min, and finally kept at 4 °C in a thermal cycle. PCR amplification products were visualized on 1% agarose electrophoresis gels stained with GoldView. Purification and sequencing of PCR products were carried out at Beijing Tsingke Biological Engineering Technology and Services Co., Ltd. (Beijing, China).

### 2.3. Phylogenetic Analysis

Related sequences were downloaded from GenBank based on BLAST searches and previous publications [15,19,28,29] (Table 1). Two species of *Anisomeridium* were selected as the outgroup taxa. Consensus sequences were assembled with BioEdit v. 7.2.3 [30], aligned using MAFFT v. 7.110 online programs (http://mafft.cbrc.jp/alignment/server/, accessed on 16 August 2021) [31] and manually edited via BioEdit v. 7.2.3 [30]. 

The evolutionary model of nucleotide substitution was selected independently for each locus using MrModeltest 2.2 [32]. GTR + G + I substitution model was selected for LSU, SSU and *tef1-α* partitions. The analyses of maximum likelihood (ML), maximum parsimony (MP) and Bayesian inference (BI) were carried out as detailed in Dissanayake et al. [33]. The programs used were raxmlGUI v. 1.3 [34] for ML, PAUP v.4.0b10 [35] for MP and MrBayes v3.1.2 [36] for BI. The alignment generated in this study was submitted to TreeBASE (https://treebase.org/treebase-web/home.html, accessed on 30 November 2021) with a submission number: ID29068. Taxonomic novelties were submitted to the Faces of Fungi database (https://www.facesoffungi.org/, accessed on 25 November 2021) [37] and MycoBank (https://www.mycobank.org/, accessed on 25 November 2021) [38]. Trees were visualized with FigTree v. 1.4.2 [39], and the layout was edited using Adobe Illustrator CS6. 

**Table 1 jof-08-00076-t001:** Taxa used in the phylogenetic analyses and their GenBank accession numbers.

Taxa	Strain Numbers	GenBank Accession Numbers			References
		LSU	SSU	*tef1-α*	
*Acrospermum adeanum*	M 133	EU940104	EU940031	N/A	[40]
*A. compressum*	M 151	EU940084	EU940012	N/A	[40]
*A. graminum*	M 152	EU940085	EU940013	N/A	[40]
*Anisomeridium phaeospermum*	MPN539	JN887394	JN887374	JN887418	[41]
*A. ubianum*	MPN94	N/A	JN887379	JN887421	[41]
*Melomastia chromolaenae*	MFLUCC 17–1434 *	KY111905	MT214413	MT235800	[42]
*M. clematidis*	MFLUCC 17–2092 *	MT214607	MT226718	MT394663	[19]
*M. distoseptata*	NFCCI: 4377 *	MH971236	N/A	N/A	[15]
*M. fulvicomae*	MFLUCC 17–2083 *	MT214608	MT226719	N/A	[19]
** *M. fusispora* **	**CGMCC 3.20618 ***	**OK623464**	**OK623494**	**OL335189**	**This study**
** *M. fusispora* **	**UESTCC 21.0001**	**OK623465**	**OK623495**	**OL335190**	**This study**
*M. italica*	MFLUCC 15–0160 *	MG029458	MG029459	N/A	[18]
*M. maolanensis*	GZCC 16–0102 *	KY111905	KY111906	KY814762	[28]
*M. neothailandica*	MFLU 17–2589 *	NG068294	N/A	N/A	[43]
** *M. oleae* **	**CGMCC 3.20619 ***	**OK623466**	**OK623496**	**OL335191**	**This study**
** *M. oleae* **	**UESTCC 21.0003**	**OK623467**	**OK623497**	**OL335192**	**This study**
** *M. oleae* **	**UESTCC 21.0005**	**OK623468**	**OK623498**	**OL335193**	**This study**
** *M. oleae* **	**UESTCC 21.0006**	**N/A**	**OK623499**	**OL335194**	**This study**
*M. phetchaburiensis*	MFLUCC 15-0951 *	MF615402	MF615403	N/A	[44]
*M. rhizophorae*	BCC 15481	N/A	KF160009	N/A	[20]
*M*. *rhizophorae*	JK 5439 A	GU479799	GU479766	GU479860	[20]
** *M. sichuanensis* **	**CGMCC 3.20620 ***	**OK623469**	**OK623500**	**OL335195**	**This study**
** *M. sichuanensis* **	**UESTCC 21.0008**	**OK623470**	**OK623501**	**OL335196**	**This study**
*M. sinensis*	MFLUCC 17–1344 *	MG836699	MG836700	N/A	[45]
*M. thailandica*	MFLUCC 15–0945 *	KX611366	KX611367	N/A	[46]
*M. thamplaensis*	MFLUCC 15–0635 *	KX925435	KX925436	KY814763	[28]
*M. tiomanensis*	MFLUCC 13–0440 *	KC692156	KC692155	KC692157	[20]
** *M. winteri* **	**CGMCC 3.20621 ***	**OK623471**	**OK623502**	**OL335197**	**This study**
*Muyocopron castanopsis*	MFLUCC 14–1108 *	KU726965	KU726968	MT136753	[47]
*M. dipterocarpi*	MFLU 17–2608	KU726966	KU726969	MT136754	[47]
*M. garethjonesii*	MFLUCC 16–2664 *	KY070274	KY070275	N/A	[48]
*M. heveae*	MFLUCC 17–0066 *	MH986832	MH986828	N/A	[29]
*M. lithocarpi*	MFLUCC 14–1106 *	KU726967	KU726970	MT136755	[47]
*Palawania thailandensis*	MFLICC 14–1121 *	KY086494	N/A	N/A	[49]
*P. thailandensis*	MFLU 16–1873	KY086493	KY086495	N/A	[49]
*Stigmatodiscus enigmaticus*	CBS 132036 *	KU234108	KU234130	N/A	[50]
*S. labiatus*	CBS 144700 *	MH756065	MH756065	MH756083	[51]
*S. oculatus*	CBS 144701 *	MH756069	N/A	MH756086	[51]
*S. pruni*	CBS 142598 *	KX611110	KX611110	KX611111	[52]
*Superstratomyces albomucosus*	CBS 140270 *	KX950439	KX950467	KX950471	[53]
*S. atroviridis*	CBS 140272 *	NG058271	NG063075	LR812724	[53]
*S. flavomucosus*	CBS 353.84 *	KX950438	KX950462	KX950470	[53]

Type strains are indicated with *; newly generated sequences in this study are indicated in bold. “N/A” denotes sequences that are not available. Abbreviations: BCC: BIOTEC Culture Collection, Bangkok, Thailand; CBS: CBS-KNAW Collections, Westerdijk Fungal Biodiversity Institute, Utrecht, The Netherlands; CGMCC: China General Microbiological Culture Collection Center; MFLU: Mae Fah Luang University Herbarium Collection; MFLUCC: Mae Fah Luang University Culture Collection, Chiang Rai, Thailand; MPN: Matthew P. Nelsen; NFCCI: National Fungal Culture Collection of India, India; GZCC: Guizhou culture collection, Guizhou, China; JK: J. Kohlmeyer; UESTCC: University of Electronic Science and Technology Culture Collection.

## 3. Results

### 3.1. Phylogenetic Analyses

The aligned sequence matrix comprises LSU, SSU, and *tef1-α* sequence data of 42 taxa representing four orders (*Acrospermales*, *Muyocopronales*, *Stigmatodiscales* and *Superstratomycetales*) and two families (*Palawaniaceae* and *Pleurotremataceae*), with *Anisomeridium phaeospermum* (MPN539) and *A. ubianum* (MPN94) as outgroup taxa. The aligned dataset comprised 2899 characters, including gaps (LSU: 1–894; SSU: 895–1942; *tef1-α*: 1943–2899). The best scoring RAxML tree is shown in Figure 1. The analyzed ML, MP and Bayesian trees were similar in topology and did not conflict significantly. *Pleurotremataceae* was resolved as a monophyletic clade. Nine strains obtained in this study were grouped in *Pleurotremataceae* and represented four new species *viz*. *Melomastia fusispora, M. oleae*, *M. sichuanensis* and *M. winteri*. Two isolates of *Melomastia sichuanensis* (CGMCC3.20620, UESTCC 21.0008) are sisters to *M. clematidis* (MFLUCC 17-2092) with high statistical support (92% ML BS/86% MP BS/ 1.0 PP). Four isolates of *Melomastia oleae* (UESTCC 21.0006, UESTCC 21.0005, CGMCC3.20619, UESTCC 21.0003) formed a distinct clade sister to the clade containing *M. fusispora* (CGMCC3.20618, UESTCC 21.0001), *M. winteri* (CGMCC3.20621) and *M. thamplaensis* (MFLUCC 15-0635) with strong statistical support (100% ML BS/100% MP BS/ 1.0 PP). The two isolates of *Melomastia fusispora* (CGMCC3.20618, UESTCC 21.0001) clustered with *M. winteri* (CGMCC3.20621) with high statistical support (97% ML BS/100% MP BS/1.0 PP).

### 3.2. Taxonomy

#### 3.2.1. Melomastia

***Melomastia*** Nitschke ex Sacc., Atti Soc. Veneto-Trent. Sci. Nat., Padova, Sér. 44: 90 (1875), emend.

MycoBank: MB 3118.

=*Dyfrolomyces* K.D. Hyde, K.L. Pang, Alias, Suetrong & E.B.G. Jones, Cryptogamie, Mycologie, 34 (1): 223–232 (2013).

Saprobic on woody branches, twigs and culms in terrestrial, freshwater, and mangrove habitats. Sexual morph: appearing as raised, dome-shaped, black dots on host surface. *Ascomata*: perithecial, scattered, solitary, immersed, erumpent or rarely superficial, globose, subglobose or ellipsoidal, coriaceous to carbonaceous, dark brown, clypeate, papillate ostiole. *Papilla*: narrow, conical, periphysate, often umbilicate. *Periphyses*: hyaline, filamentous. *Peridium*: one-layered or two-layered, with an outer layer composed of host cells, interdispersed with fungi hyphae, forming a *textura intricata* and inner layer of thick-walled cells of *textura angularis*. *Hamathecium*: comprising numerous, filamentous, flexuose, septate pseudoparaphyses embedded in a gelatinous matrix. *Asci*: 8-spored, bitunicate, fissitunicate, cylindrical, with a relatively short pedicel, apically rounded or flattened with a distinct ocular chamber and ring-like subapical apparatus. *Ascospores*: overlapping uni-seriate, hyaline, ellipsoid to fusiform, rarely curved, 1–10-septate, not or slightly constricted at the septum, with or without a mucilaginous sheath. Asexual morph: unknown. 

Type species: *Melomastia mastoidea* (Fr.) J. Schröt., in Cohn, Krypt.-Fl. Schlesien (Breslau) 3.2(3): 320 (1894).

MycoBank: 205357.

≡*Sphaeria lonicerae* Sowerby, Col. Engl. Fung. Mushr. (London) 3(no. 27) (1803).

*=Sphaeria mastoidea* Fr., K. svenska Vetensk-Akad. Handl., Ser. 338: 267 (1817).

*=Sphaeria revelata* Berk. & Broome, Ann. Mag. nat. Hist., Ser. 29: 325 (1852).

*=Sphaeria fraxinicola* Curr., Trans. Linn. Soc. London 22: 333 (1859).

*=Melomastia friesii* Nitschke, in Fuckel, Jb. nassau. Ver. Naturk. 25–26: 306 (1871).

*=Conisphaeria mastoidea* (Fr.) Stev., (1879).

*=Psilosphaeria lonicerae* (Sowerby) Stev., Mycol. Scot.: 388 (1879).

*=Phyllosticta fraxinicola* Sacc., Syll. fung. (Abellini) 3: 12 (1884).

*=Trematosphaeria mastoidea* (Fr.) G. Winter, Rabenh. Krypt.-Fl., Edn 2 (Leipzig) 1.2: 274 (1885).

*=Conisphaeria friesii* (Nitschke) Cooke, Grevillea 16 (no. 79): 87 (1888).

*=Hypoxylon mastoideum* (Fr.) P.M.D. Martin, Jl S. Afr. Bot. 34: 176 (1968).

Notes: *Melomastia* was introduced by Nitschke [16] with *M. friesii* as the type species found on *Viburnum opulus* L. (*Adoxaceae*) in Germany. Later, the species was synonymized under *Melomastia mastoidea* [54] and is recognized by its immersed, obpyriform ascomata with a conical papillate ostiole, peridium of several layers of brown to dark brown, compressed cells, unitunicate asci and 2-septate, oblong, obtuse ends and hyaline ascospores [2]. Chen et al. [55] checked the paratype of *Melomastia mastoidea* isolated from Taiwan and noted its ascomatal peridium composed of a 2-zone. Among currently known *Melomastia* species, *M. clematidis, M. fulvicomae* and *M. maolanensis* also possesses a 2-zone ascomatal peridium, while *M. italica* possesses 1-zone. *Melomastia mastoidea* is saprobic on a wide range of woody and semi-woody substrata, including stems and twigs of *Epilobium*, *Eupatorium*, *Hedera*, *Lonicera*, *Symphoricarpus* and *Uimus* [2].

Pang et al. [20] introduced *Dyfrolomyces* to accommodate species with immersed coriaceous ascomata with clypeated ostiolar neck, peridium composed of 2-zone, multicellular ascospores. The family *Dyfrolomycetaceae* was proposed based on the phylogenetic analyses of a multi-gene [20] to accommodate this single genus. Morphologically similar *Melomastia* species were not included in their analyses due to the limited availability of the sequence data, and thus the affiliation of these two genera has remained unclear.

The presence of multi-septate ascospores in “*Dyfrolomyces*” is key in differentiating “*Dyfrolomyces*” from *Melomastia* species. However, some *Melomastia* species, such as *M. maolanensis* has 3-septate ascospores, which are morphologically similar to “*Dyfrolomyces*” but were phylogenetically related to *Melomastia* in previous studies [15,18,20]. Thus, the ascospore septation is not a specific character for the generic delimitation of these two genera. In the present study, phylogenetic analyses of combined LSU, SSU and *tef1-α* sequence data showed taxa of *Pleurotremataceae* were separated into two major clades with well-supported bootstrap values (100% ML BS/78% MP BS/1.00 BI, Figure 1). The clade A consisted of *Melomastia rhizophorae* (“*Dyfrolomyces rhizophorae*”), *M. thailandica* (“*D. thailandica*”), *M. neothailandica (*“*D*. *neothailandicus*”), *M. phetchaburiensis* (“*D. phetchaburiensis*”), *M. fulvicomae*, *M. sichuanensis*, *M. clematidis*, *M. italica*, *M. chromolaenae* (“*D. chromolaenae*”) and *M. tiomanensis* (“*D. tiomanensis*”), while the clade B consisted of *M. oleae*, *M. fusispora*, *M. winteri*, *M. thamplaensis* (“*D. thamplaensis*”), *M. sinensis* (“*D. sinensis*”) and *M. distoseptata* (“*D. distoseptatus*”). However, the taxa in these two clades have no noticeable morphological differences. We conclude the *Melomastia* and “*Dyfrolomyces*” are congeneric based on molecular phylogeny and morphology. Further studies with fresh collections are needed to resolve the taxonomic relationships in *Pleurotremataceae* and its sexual–asexual connections. 

***Melomastia fusispora*** W.L. Li, Maharachch. & Jian K. Liu, **sp. nov.** Figure 2.

MycoBank number: MB 841499; Facesoffungi number: FoF10533.

**Etymology:** Refers to the shape of the ascospore.

Holotype: HKAS 121316.

*Saprobic* on dead branches of *Olea europaea.*
**Sexual morph:**
*Ascomata*, visible as numerous, black, cone-shaped structures on the host surface, solitary, gregarious, immersed to erumpent through host tissue, pyriform, coriaceous to carbonaceous, dark brown to black, rough-walled, ostiolate, 432–624 × 527–618 μm diam. ($\bar{x}$ = 528 × 572 μm, n = 15). *Ostioles* central, carbonaceous, dark brown to black, papillate, periphyses filling the ostiolar canal 19.14–343.5 × 65.8–283.5 μm ($\bar{x}$ = 174.5 × 181 μm, n = 15). *Peridium* two-layered, an outer layer of cells of *textura intricata* composed of host cells interspersed with fungal hyphae and an inner layer of thick-walled cells of *textura angularis*, 25.5–61.5 μm ($\bar{x}$ = 41 μm, n = 15) wide. *Hamathecium* comprises numerous, dense, filiform, unbranched, hyaline, aseptate pseudoparaphyses, 2–2.6 μm ($\bar{x}$ = 2.3 μm, n = 30) wide. *Asci* 8-spored, bitunicate, cylindrical, slightly flexuous, apically round, with well-developed ocular chamber, 200–231 × 7.6–9.2 μm ($\bar{x}$ = 215 × 8.4 μm, n = 30), cylindrical pedicellate 11–17.5 × 3.5–4.3 μm ($\bar{x}$ = 14 × 3.9 μm, n = 30). *Ascospores* uniseriate, partially overlapping, hyaline, fusiform, with rounded to acute ends, narrow towards apex, 3-septate, constricted at the central septum, with guttules in each cell, surrounded by an irregular and thin gelatinous sheath, 27.5–32 × 6.5–7.5 μm ($\bar{x}$ = 30 × 7 μm, n = 30). **Asexual morph:** Undetermined. 

**Culture characteristics:** Colonies on PDA reaching 40 mm diam. after 4 weeks at 25 °C. Culture from above, brownish grey, forming zonate grey, fluffy mycelium at the edge; reverse dark brown, greyish orange at the edge.

**Material examined:** China, Sichuan Province, Chengdu City, Shuangliu District, Olive Base, 30°33.25′ N, 103°99.62′ E, at an altitude of 438 m (mountainside), on dead branch of *Olea europaea*, 27 March 2021, W.L. Li, GL 139 (HKAS 121316, holotype; HUEST 21.0002, isotype), ex-type living culture CGMCC3.20618 = UESTCC 21.0002. Additional genes sequenced: ITS OK623480, RPB2 OK632627; *ibid.*, T. Zhang, GL 136 (HUEST 21.0001, paratype), living culture UESTCC 21.0001. Additional genes sequenced: ITS OK623481, RPB2 OK632628. 

**Notes:** Phylogenetic analysis showed that *Melomastia fusispora* is sister to *M. winteri* with high support (97% ML BS, 100% MP BS, 1.00 PP). *Melomastia fusispora* resembles *M. winteri* in having cylindrical asci and fusiform, 3-septate ascospores. However, *M. fusispora* has larger asci (200–231 vs. 165–189 μm), pedicel (11–17.5 vs. 4.8–6.5 μm) and ascospore (27.5–32 vs. 25–30 μm) when compared to *M. winteri.* In addition, the ascospore ends of the *Melomastia winteri* are usually pointed, which differs from *M. fusispora* as the latter’s ascospore has round to acute ends. A comparison of the 914 nucleotides across the *rpb2* gene region of *M. fusispora* (CGMCC3.20618) and *M. winteri* (CGMCC3.20621) reveals 39 bp (base pair) differences (4.26%).

**Figure 2 jof-08-00076-f002:**
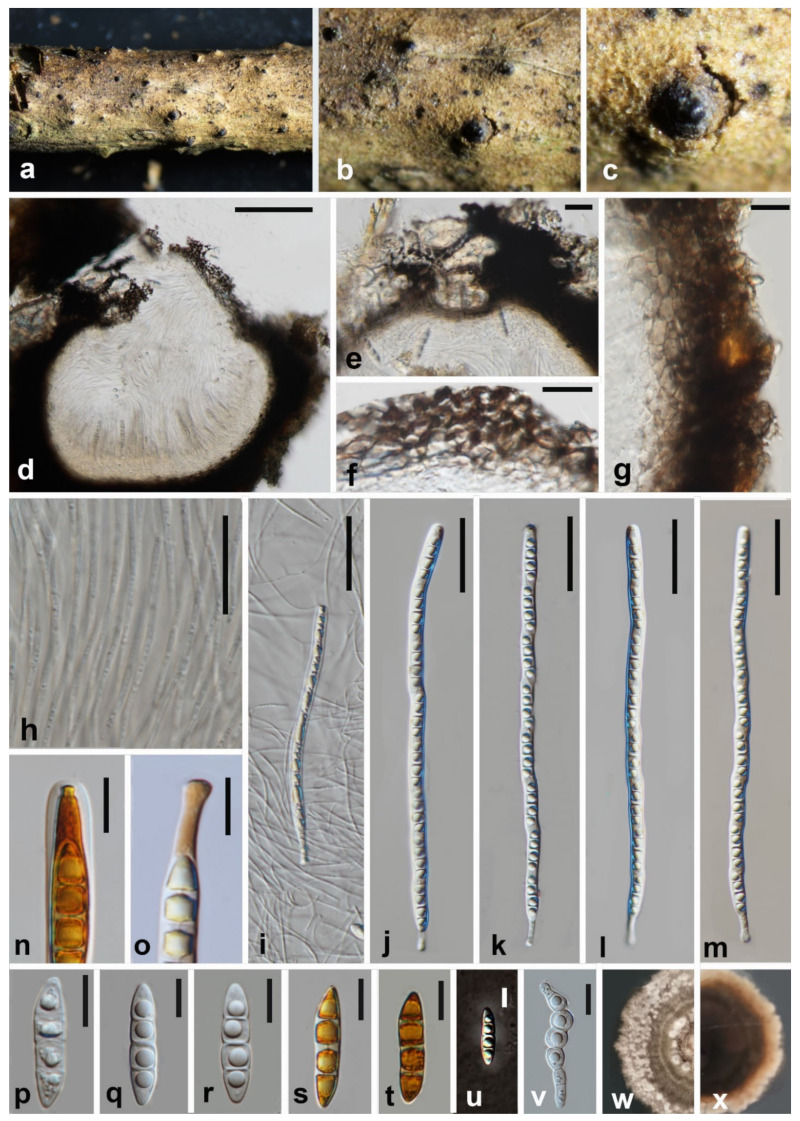
***Melomastia fusispora*** (HKAS 121316, holotype). (**a**–**c**) Ascomata on the substrate. (**d**) Vertical section of the ascomata. (**e**) Vertical section of the ostiole. (**f**,**g**) Peridium (**h**) Pseudoparaphyses. (**i**–**m**) Asci. (**n**) Ocular chamber in Lugol’s iodine. (**o**) Pedicel in Lugol’s iodine. (**p**–**u**) Ascospora. ((**s**,**t**) in Lugol’s iodine, (**u**) in India ink). (**v**) Germinating ascospore. (**w**) Upper view of the colony on PDA after 14d. (**x**) Reverse view of the colony on PDA after 14d. Scale bars: (**d**) = 100 μm, (**e**,**g**) = 20 μm, (**f**,**n**–**u**) = 10 μm, (**h**–**m**) = 40 μm.

***Melomastia oleae*** W.L. Li, Maharachch. & Jian K. Liu, **sp. nov.**
Figure 3.

MycoBank number: MB 841500; Facesoffungi number: FoF10534.

**Etymology:** The species epithet refers to the host genus on which it was collected.

**Holotype:** HKAS 121315.

*Saprobic* on dead branches of *Olea europaea. **Sexual morph***: *Ascomata* visible as numerous, black, cone-shaped structures on host surface, solitary, gregarious, semi-immersed, globose to compressed globose, carbonaceous, dark brown to black, rough-walled, ostiolate, 410–440 × 493–520 μm. ($\bar{x}$ = 425 × 506.5 μm, n = 15). *Ostioles* neck crest-like, central, carbonaceous, dark brown to black, papillate, periphyses filling the ostiolar canal, 20.5–50 × 60–83 μm ($\bar{x}$ = 35 × 72 μm, n = 15). *Peridium* two-layered, outer thick, carbonaceous and inner composed of 5–6 layers of hyaline cells of *textura angularis* to *textura prismatica*, 54–65 μm wide. *Hamathecium* composed of numerous, dense, filiform, unbranched, aseptate pseudoparaphyses, 2–2.5 μm wide. *Asci* 8-spored, bitunicate, cylindrical, slightly flexuous, apically rounded with ocular chamber, 209–237 × 7.5–9 μm ($\bar{x}$ = 223 × 8 μm, n = 30), cylindrical pedicellate 9–12 × 3.5–5.8 μm ($\bar{x}$= 10.5 × 4.6 μm). *Ascospores* uniseriate, partial overlapping, fusiform with obtuse ends, hyaline, 3-septate, slightly constricted at the septa, with guttules in each cell, 28–34 × 6–7 μm ($\bar{x}$ = 31 × 6.5 μm, n = 30). **Asexual morph:** Undetermined. 

**Culture characteristics:** Colonies on PDA reaching 15 mm diam. after 2 weeks at 25 °C. Cultures from above white, dense, circular, margin entire, papillate; reverse dark brown. 

**Material examined:** China, Sichuan Province, Chengdu City, Shuangliu District, Olive Base, 30°33.25′ N, 103°99.62′ E, at an altitude of 432 m (the foot of mountain), 30 January 2021, on dead branch of *Olea europaea*, W.L. Li, GL 031 (HKAS 121315, holotype; HUEST 21.0004, isotype), ex-type living culture CGMCC3.20619 = UESTCC 21.0004. Additional genes sequenced: ITS OK623482; *ibid.*, T. Zhang, GL 004 (HUEST 21.0003, paratype), living culture UESTCC 21.0003; *ibid.*, at an altitude of 438 m (Mountain side), 27 March 2021, on dead branch of *Olea europaea*, W.L. Li, GL 128 (HUEST 21.0005, paratype isotype), living culture UESTCC 21.0005. Additional genes sequenced: ITS OK623483; *ibid.*, Z.P. Liu, GL 135 (HUEST 21.0006, paratype), living culture UESTCC 21.0006. Additional genes sequenced: ITS OK623484.

**Notes:** Phylogenetic analysis results showed that *Melomastia oleae* clustered with *M. fusispora*, *M. thamplaensis* and *M. winteri*, but represented a distinct clade with high support (100% ML BS, 100% MP BS, 1.00 PP). *Melomastia oleae* share similar ascus and ascospore morphology with *M. fusispora*, *M. thamplaensis* and *M. winteri*. However, *M. winteri* and *M. thamplaensis* have shorter asci than *M. oleae* (165–189 μm vs. 114–160 μm vs. 209–237 μm)*. Melomastia fusispora* is easily differentiated from *M. oleae* by having non-carbonised peridium and relatively lager ascomata (528 × 572 μm vs. 425 × 506.5 μm). Additionally, *tef1-α* sequence comparison reveals 48 bp differences without gaps across 875 nucleotides (5.48%) between *Melomastia fusispora* and *M. oleae*.

**Figure 3 jof-08-00076-f003:**
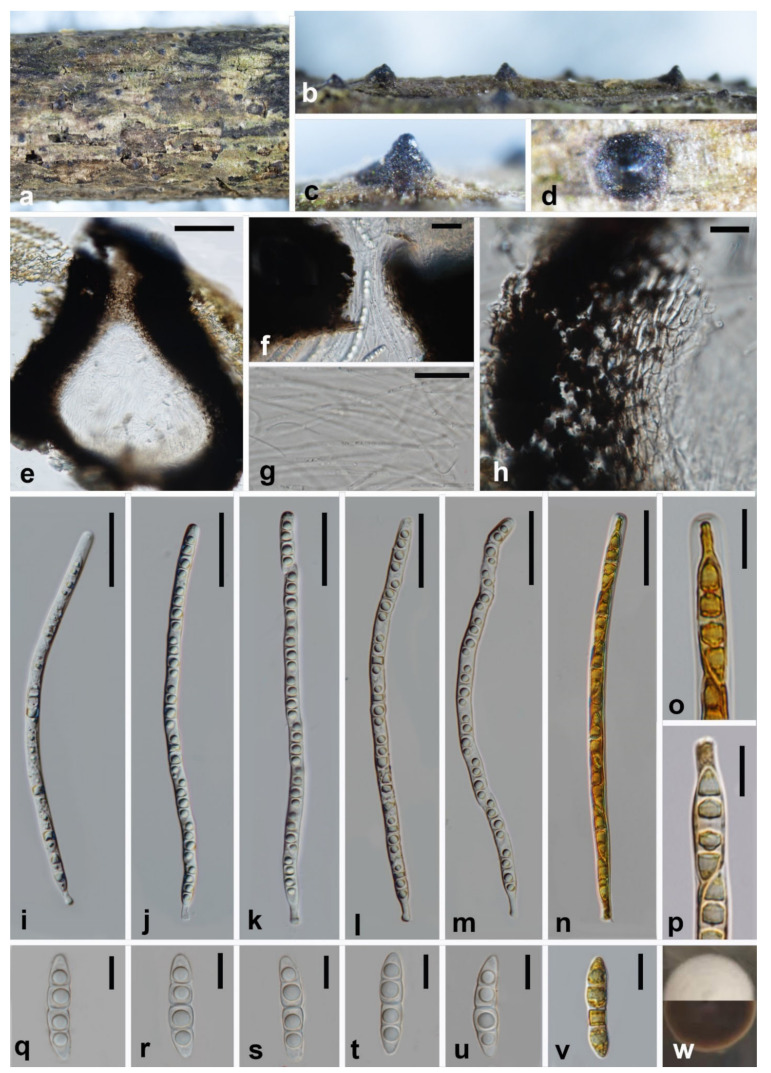
***Melomastia oleae*** (HKAS 121315, holotype). (**a**–**d**) Ascomata on the substrate. (**e**) Verticle section of the ascoma. (**f**) Vertical section of the ostiole. (**g**) Pseudoparaphyses. (**h**) Peridium. (**i**–**n**) Asci ((**n**) in Lugol’s iodine). (**o**) ocular chamber in Lugol’s iodine. (**p**) pedicel in Lugol’s iodine. (**q**–**v**) Ascospora ((**v**) in Lugol’s iodine). (**w**) Upper and reverse view of the colony on PDA after 14d. Scale bars: (**e**) = 100 μm, (**f**) = 20 μm, (**g**,**h**,**o**–**v**) = 10 μm, (**i**–**n**) = 40 μm.

***Melomastia sichuanensis*** W.L. Li, Maharachch. and Jian K. Liu, **sp. nov**. Figure 4.

MycoBank number: MB841501; Facesoffungi number: FoF10535.

**Etymology:** Refers to the place where the type species was collected.

**Holotype:** HKAS 121313.

*Saprobic* on dead branches of *Olea europaea.*
**Sexual morph:**
*Ascomata* only ostioles visible at the surface of host, solitary, gregarious, immersed, globose to subglobose, pseudothecia, coriaceous to carbonaceous, dark brown to black, rough-walled, ostiolate, 419.5–506 × 335–577 μm ($\bar{x}$ = 462.5 × 456 μm, n = 15). *Ostioles* central, carbonaceous, dark brown to black, papillate, periphyses filling the ostiolar canal, 90–120 × 85–110 μm ($\bar{x}$ = 105 × 97.5 μm, n = 15). *Clypeus* black, thick-walled. *Peridium* two-layered, an outer layer of cells of *textura intricata* composed of host cells interspersed with fungal hyphae, carbonaceous and the inner layer comprising lightly brown cells and thin, 44–85.5 μm ($\bar{x}$ = 65 μm, n = 15) wide. *Hamathecium* composed of numerous, dense, filiform, unbranched, septate, cellular pseudoparaphyses, 2.5–3.7 μm ($\bar{x}$ = 3.1 μm, n = 15) wide. *Asci* 8-spored, bitunicate, cylindrical, apically round, with a small ocular chamber, 101–112.5 × 6.5–7.6 μm ($\bar{x}$ = 107 × 7 μm, n = 30), cylindrical pedicellate 3.5–7 × 3.2–3.8 μm ($\bar{x}$ = 5.3 × 3.5 μm, n = 30). *Ascospores* uniseriate, partial overlapping, broad fusiform with rounded ends, hyaline, 3-septate, constricted at the septa, with guttules in each cell, smooth-walled with mucilaginous sheath, 15–17.5 × 4.7–5.1 μm ($\bar{x}$ = 16 × 5 μm, n = 30). ***Asexual morph***: Undetermined. 

**Culture characteristics.** Colonies on PDA reaching 30 mm diam. after 4 weeks at 25 °C. Culture from above, yellowish radiating outwardly, with yellow ring in the middle, medium dense, circular shape, covered with cream mycelium; reverse: brown in the middle with an orange ring, dark yellow radiating outwardly.

**Material examined:** China, Sichuan Province, Chengdu City, Shuangliu District, Olive Base, 30°33.25′ N, 103°99.62′ E, at an altitude of 432 m (the foot of mountain), 30 January 2021, W.L. Li, GL 001 (HKAS 121313, holotype; HUEST 21.0007, isotype), ex-type living culture CGMCC3.20620 = UESTCC 21.0007. Additional genes sequenced: RPB2 OK632629; *ibid.*, Z.P. Liu, GL 005 (HUEST 21.0008, paratype), living culture, UESTCC 21.0008. Additional genes sequenced: RPB2 OK632630.

**Notes:** Phylogenetic analysis results showed that *Melomastia sichuanensis* formed a moderately supported clade sister to *M. clematidis* (92% ML BS, 86% MP BS, 1.00 BI). In addition, the comparison of *tef1-α* sequence data between the two species showed 46 nucleotides differences without gaps across the 878 nucleotides (5.24%), but only 1 bp difference across 829 nucleotides of LSU and 2 bp differences across the 766 nucleotides of SSU. Morphologically, *M. sichuanensis* resembles *M. clematidis* in having 3-septate ascospore with a thin mucilaginous sheath, but *M. sichuanensis* can be distinguished from *M. clematidis* in its shorter asci (101–112.5 μm vs. 115–160 μm). In addition, *M. clematidis* has ellipsoidal to fusiform ascospores with acute ends, while they are broad fusiform with rounded ends in *M. sichuanensis* [19].

**Figure 4 jof-08-00076-f004:**
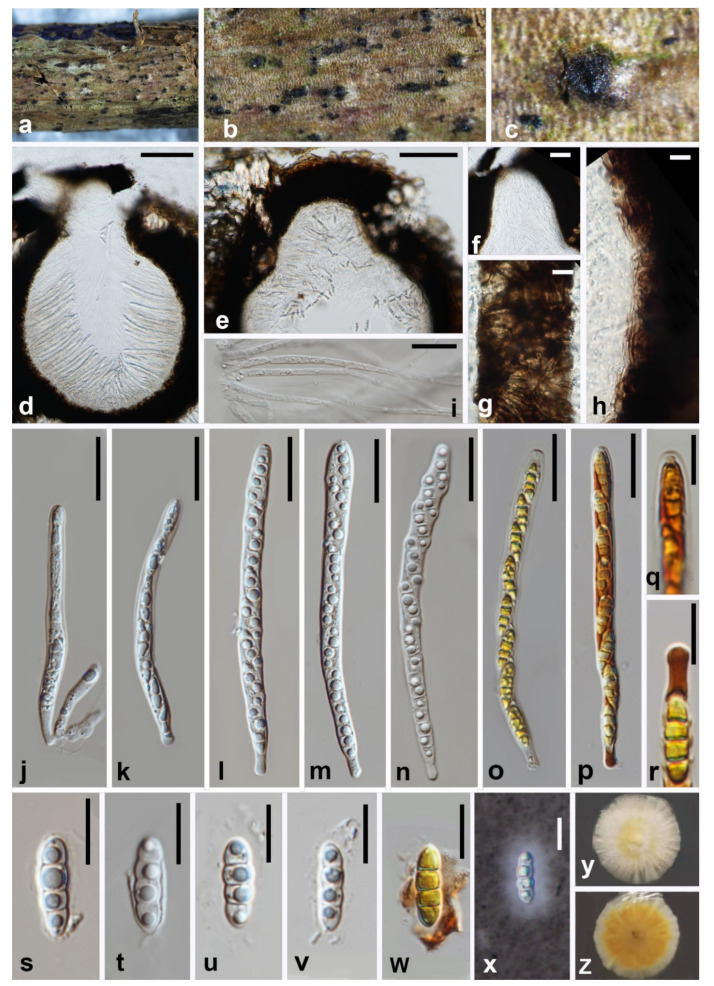
***Melomastia sichuanensis*** (HKAS 121313, holotype). (**a**–**c**) Ascomata on the substrate. (**d**,**e**) Vertical section of ascoma. (**f**) Vertical section of ostiole. (**g,h**) Peridium. (**i**) Pseudoparaphyses. (**j**–**p**) Asci ((**o**,**p**) in Lugol’s iodine). (**q**) ocular chamber in Lugol’s iodine. (**r**) pedicel in Lugol’s iodine. (**s**–**x**) Ascospora ((**w**) in Lugol’s iodine, **x** in India ink). (**y**,**z**) Upper and reverse view of the colony on PDA after 14d. Scale bars: (**d**,**e**) = 100 μm, (**f**) = 40 μm, (**g**,**h**,**q**–**x**) = 10 μm, (**i**–**p**) = 20 μm.

***Melomastia winteri*** W.L. Li, Maharachch. & Jian K. Liu, **sp. nov.**
Figure 5.

MycoBank number: MB841502; Facesoffungi number: FoF10536.

**Etymology:** The epithet “*winteri*” refers to the season when the fungus was collected.

Holotype: HKAS 121314.

Saprobic on dead branches of *Olea europaea*. **Sexual morph**
*Ascomata* solitary, gregarious, semi-immersed to immersed, globose, coriaceous to carbonaceous, dark brown to black, ostiolate, 340–365 × 364–410 μm ($\bar{x}$ = 352 × 387 μm, n = 15). *Ostioles* central, coriaceous to carbonaceous, dark brown to black, papillate, periphyses filling the ostiolar canal, 105–108 × 106–113 μm ($\bar{x}$ = 106.5 × 109.5 μm, n = 10). *Peridium* two-layered, outer thick, carbonaceous and inner composed of 3–4 layers of hyaline to lightly brown cells of *textura angularis* to *textura prismatica*, 55–62.5 μm ($\bar{x}$= 59 μm, n = 10) wide. *Hamathecium* composed of numerous, dense, filiform, unbranched, septate, cellular pseudoparaphyses, 1.5–3.5 μm wide. *Asci* 8-spored, bitunicate, cylindrical, slightly flexuous, apically round, with a distinct ocular chamber, 165–189 × 7–8.5 μm ($\bar{x}$ = 177 × 7.5 μm, n = 30), cylindrical pedicellate 4.8–6.5 × 3.5–4.6 μm ($\bar{x}$ = 5.6 × 4 μm, n = 30). *Ascospores* uniseriate, partially overlapping, fusiform with acute ends, hyaline, 3-septate, deeply constricted at the median septum, with guttules in each cell, lacking a mucilaginous sheath, 25–30 × 5–6.5 μm ($\bar{x}$ = 27.5 × 6 μm, n = 30). **Asexual morph**: Undetermined. 

**Culture characteristics:** Colonies on PDA reaching 30 mm diam. after 4 weeks at 25 °C. Cultures from above, white, dense, circular, margin erose, umbonate, papillate with somewhat fluffy, pale orange at the center; reverse dark brown at the center, pale yellow at the edge. 

**Material examined:** China, Sichuan Province, Chengdu City, Shuangliu District, Olive Base, 30°33.25′ N, 103°99.62′ E, at an altitude of 432 m (the foot of the mountain), 29 January 2021, W.L. Li, GL 010 (HKAS 121314, holotype; HUEST 21.0009, isotype); extype living culture CGMCC3.20621 = UESTCC 21.0009. Additional genes sequenced: ITS OK623485, RPB2 OK632631.

**Notes:** Phylogenetic analysis showed that *Melomastia winteri* clustered with *M. fusispora* (97% ML BS, 100% MP BS, 1.00 BI). Morphologically, *M. winteri is* similar to *M. fusispora* in having 3-septate fusiform ascospore. However, *M. fusispora* can be distinguished from *M. winteri* by its larger ascomata (432–624 × 527–618 µm vs. 340–365 × 364–410 μm) and asci (200–231 µm vs. 165–189 µm). Additionally, *Melomastia fusispora* has a thin gelatinous sheath around the ascospores, which is not observed in *M. winteri*.

**Figure 5 jof-08-00076-f005:**
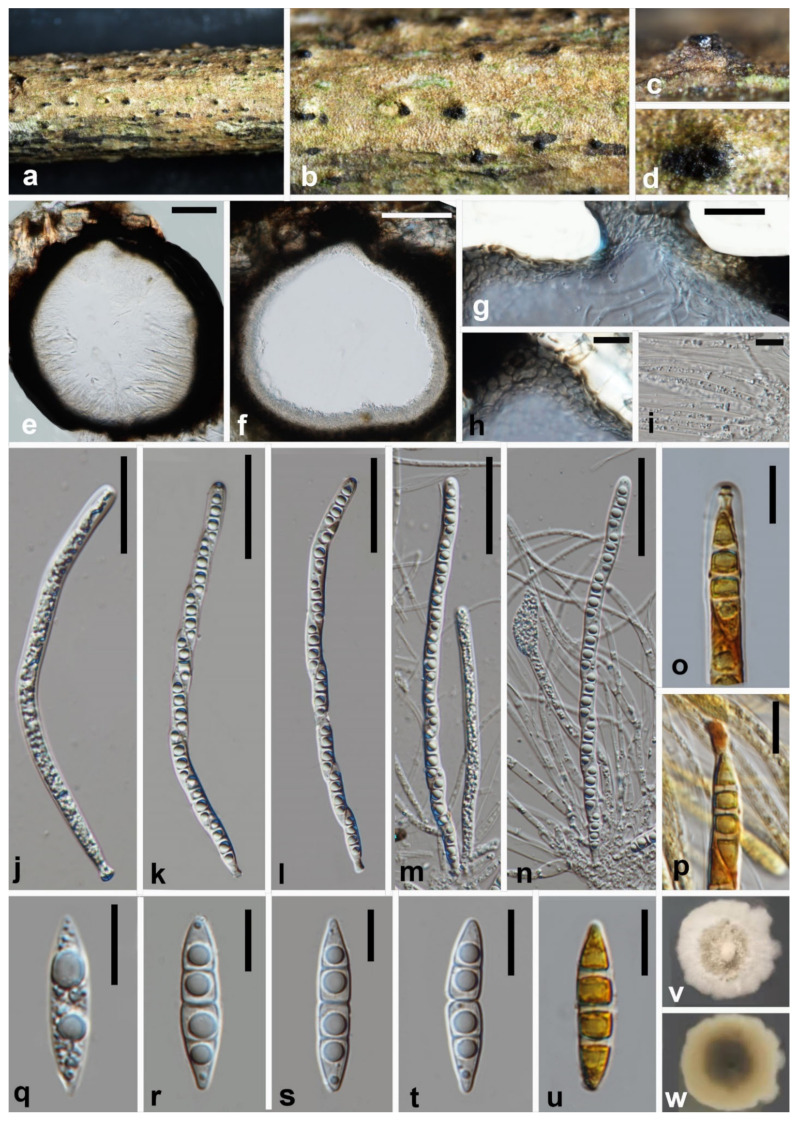
***Melomastia winteri*** (HKAS 121314, holotype). (**a**–**d**) Ascomata on the substrate. (**e**,**f**) Vertical of the ascoma. (**g**) Vertical section of the ostiole. (**h**) Peridium. (**i**) Pseudoparaphyses. (**j**–**n**) Asci. (**o**) Ocular chamber in Lugol’s iodine. (**p**) Pedicel in Lugol’s iodine. (**q**–**u**) Ascospora ((**u**) in Lugol’s iodine). (**v**,**w**) Upper and reverse view of the colony on PDA after 14d. Scale bars: (**e**,**f**) = 100 μm, (**g**,**h**,**i**,**o**–**u**) = 10 μm, (**j**–**n**) = 40 μm.

#### 3.2.2. New Combination

***Melomastia aquatica*** (K. M. Tsui, K.D. Hyde, I.J. Hodgkiss and Goh) W.L. Li, Maharachch. & Jian K. Liu, comb. nov.

≡*Saccardoella aquatica* K.M. Tsui, K.D. Hyde, I.J. Hodgkiss, T.K. Goh, Mycologia 90(4): 701, 1998. 

*=Dyfrolomyces aquaticus* (K.M. Tsui, K.D. Hyde, I.J. Hodgkiss, T.K. Goh) J.F. Zhang, J.K. Liu, K.D. Hyde & Z.Y. Liu, Phytotaxa 313 (3): 267–277, 2017.

MycoBank: MB 842085.

***Melomastia chromolaenae*** (Mapook & K.D. Hyde) W.L. Li, Maharachch. & Jian K. Liu, comb. nov.

≡*Dyfrolomyces chromolaenae* Mapook & K.D. Hyde, Fungal Diversity 101: 1–175, 2020.

MycoBank: MB 842086.

***Melomastia distoseptata*** (M. Niranjan & V.V Sarma) W.L. Li, Maharachch. & Jian K. Liu, comb. nov.

≡*Dyfrolomyces distoseptatus* M. Niranjan & V.V Sarma, Fungal Diversity 105: 17–318, 2020.

MycoBank: MB 842087.

***Melomastia******mangrovei*** (K.D. Hyde) W.L. Li, Maharachch. & Jian K. Liu, comb. nov.

≡*Saccardoella mangrovei* K.D. Hyde, Mycologia, 84(5), 806, 1992.

=*Dyfrolomyces mangrovei* (K.D. Hyde) K.D. Hyde, K.L. Pang, Alias, Suetrong & E.B.G. Jones, Cryptogamie, Mycologie, 2017, 38 (4): 507–525.

MycoBank: MB 842088.

***Melomastia******marinospora*** (K.D. Hyde) W.L. Li, Maharachch. & Jian K. Liu, comb. nov.

≡*Saccardoella **marinospora*** K.D. Hyde, Mycologia 84(5): 803–806, 1992. 

=*Dyfrolomyces **marinosporus*** (K.D. Hyde) K.D. Hyde, K.L. Pang, Alias, Suetrong & E.B.G. Jones, Cryptogamie, Mycologie, 2017, 38 (4): 507–525. 

MycoBank: MB 842089.

***Melomastia neothailandica*** (Dayarathne Jones E.B.G. and K.D. Hyde) W.L. Li, Maharachch. & Jian K. Liu, comb. nov.

≡*Dyfrolomyces neothailandicus* Dayarathne Jones E.B.G. and K.D. Hyde, Mycosphere 11(1): 1–188, 2020.

MycoBank: MB 842090.

***Melomastia phetchaburiensis*** (Dayarathne, Jones E.B.G. and K.D. Hyde) W.L. Li, Maharachch. & Jian K. Liu, comb. nov.

≡*Dyfrolomyces phetchaburiensis* Dayarathne, Jones E.B.G. and K.D. Hyde, Fungal Diversity 87: 33, 2017. 

MycoBank: MB 842091.

***Melomastia rhizophorae*** (K.D. Hyde) W.L. Li, Maharachch. & Jian K. Liu, comb. nov.

≡*Saccardoella rhizophorae* K.D. Hyde, Mycologia 84(5): 806, 1992.

=*Dyfrolomyces rhizophorae* (K.D. Hyde) K.D. Hyde, K.L. Pang, Alias, Suetrong & E.B.G. Jones, Cryptogamie, Mycologie, 2017, 38 (4): 507–525.

MycoBank: MB 842092.

***Melomastia******sinensis*** (Samarak., Tennakoon and K.D. Hyde) W.L. Li, Maharachch. & Jian K. Liu, comb. nov. 

≡*Dyfrolomyces sinensis* Samarak., Tennakoon and K.D. Hyde, Mycosphere 9(2): 354, 2018.

MycoBank: MB 842093.

***Melomastia thailandica*** (Dayarathne, Jones E.B.G. and K.D. Hyde) W.L. Li, Maharachch. and Jian K. Liu, comb. nov. 

≡*Pleurotrema thailandica* Dayarathne, Jones E.B.G. and K.D. Hyde, Fungal Diversity 81: 131, 2016.

MycoBank: MB 842094. 

***Melomastia thamplaensis*** (J.F. Zhang, J.K. Liu, K.D. Hyde and Z.Y. Liu) W.L. Li, Maharachch. and Jian K. Liu, comb. nov. 

≡*Dyfrolomyces thamplaensis* J.F. Zhang, J.K. Liu, K.D. Hyde and Z.Y. Liu, Phytotaxa 313 (3): 267–277.

MycoBank: MB 842095.

***Melomastia tiomanensis*** (K.L. Pang, S.A. Alias, K.D. Hyde, Suetrong and E.B.G. Jones) W.L. Li, Maharachch. and Jian K. Liu, comb. nov. 

≡*Dyfrolomyces tiomanensis* K.L. Pang, S.A. Alias, K.D. Hyde, Suetrong and E.B.G. Jones, Cryptogamie, Mycologie, 2013, 34 (1): 223–232.

MycoBank: MB 842096.

## 4. Discussion

Species of *Melomastia* have a wide geographical distribution, e.g., Africa, China, Germany, Italy, Japan, Poland and the United States of America (California) [18], and are reported in marine, freshwater and terrestrial habitats [9,20,28]; they do not seem to have specific host preferences. Though a few species are known as saprophytes on mangrove wood (*Avicennia marina, Kandelia, Rhizophora* sp.), the rest are reported from various woody hosts (eg. *Clematis* sp., *Chromolaena odorata* and *Vitis vinifera*) [18]. This study provides the first report of *Melomastia* species reported from *Olea europaea*.

A well-resolved revision of the family *Pleurotremataceae* is challenging since the type species of the early established genera *Melomastia* and *Pleurotrema* do not possess molecular data. Norphanphoum et al. [18] proposed combining *Dyfrolomyces* and *Pleurotrema* under *Melomastia* based on similar phenotypic characteristics, and they suggested this taxonomic assumption needs to be reinforced via increasing the number of taxa with sequence data in each genus. In the present study, we have included the taxa representing all species of *Pleurotremataceae* (of which the sequences are available in GenBank) and nine pleurotrema-like taxa obtained from olive. Considering the morphological comparison (Table 2) and multi-gene phylogenetic analysis (Figure 1), *Melomastia* and *Dyfrolomyces* are congeneric, and the epithet *Melomastia* should be adopted as it is the oldest name. Further studies with more collections of *Pleurotremataceae*, especially of *Melomastia mastoidea* and *Pleurotrema polysemum* are essential for a better understanding of the taxonomic relationship of this family.

The differences between the peridium of the ascomata and ascomatal position on the substrate probably play prominent roles in differentiating some fungal groups [56,57]. However, the results of this study suggest that it may also vary depending on the condition of habitats (terrestrial or aquatic environment) and substrates (herbaceous or woody plants), e.g., *Melomastia* produce fully immersed ascomata in an aquatic habitat (*M. tiomanensis*) and superficial or semi-erumpent ascomata in a terrestrial environment (e.g., *M. fusispora,* and *M. sinensis*) [45]; the peridium (perithecia) of the ascomata on woody plants is composed of 2-zone (e.g., *M. oleae, M. sichuanensis*), while those on herbaceous plants or semi-woody were 1-zone (e.g., *M. chromolaenae*, *M. italica*) [18,42]. Thus, the ascomatal position on the substrate and the zonation of the peridium of the ascomata have no taxonomic significance for generic delimitation in *Pleurotremataceae*, and could even change depending on the host and location. 

## 5. Conclusions

This study comprises research investigating fungi associated with oil trees in Sichuan Province in China, and here we introduced four new species of *Pleurotremataceae* isolated from olive (*Olea europaea*). With detailed descriptions, *Dyfrolomyces* was synonymized under *Melomastia* based on molecular phylogeny and morphology. Olives are cultivated in many regions of the world with Mediterranean climates, such as Australia, Chile, Italy, Peru, South Africa and the USA (California and Oregon). This study represents the first discovery of *Melomastia* on the *Olea europaea* in China.

## Figures and Tables

**Figure 1 jof-08-00076-f001:**
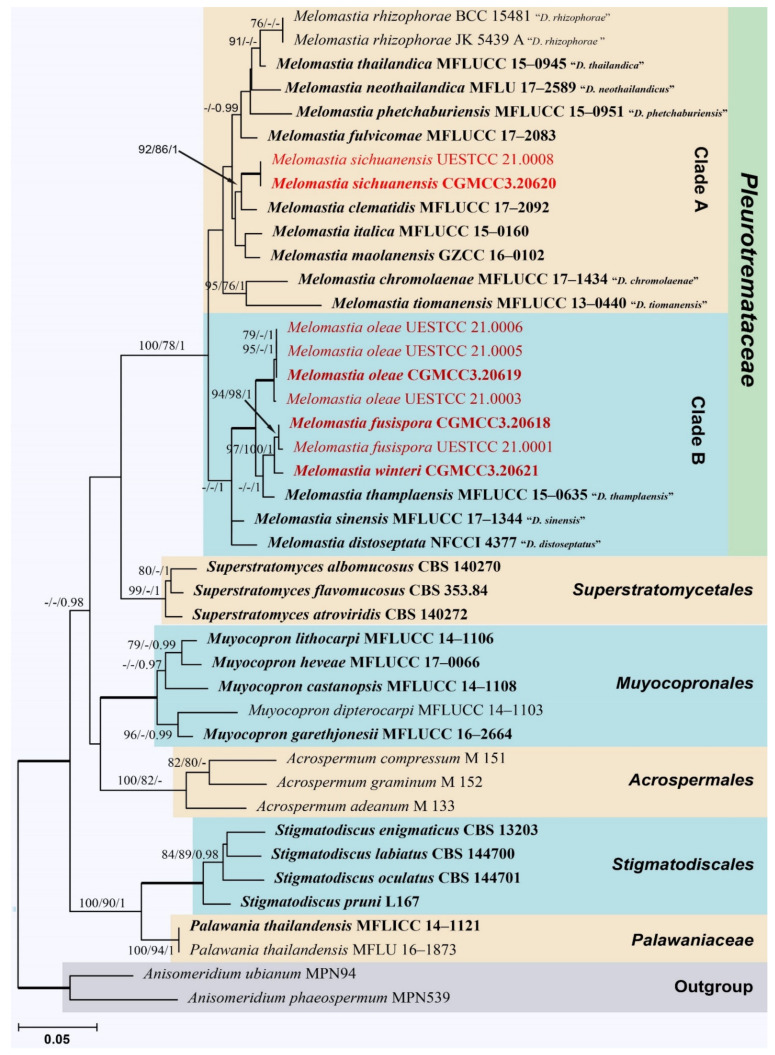
Phylogram generated from maximum likelihood (ML) based on an LSU–SSU–*tef1-α* sequence matrix. The tree is rooted with *Anisomeridium phaeospermum* (MPN539) and *A. ubianum* (MPN94). The ML, MP bootstrap supports (≥75%) and BI posterior probabilities (≥0.95 PP) supports are given near the nodes, respectively. Branches with 100% ML BS, 100% MP BS and 1.00 PP are thickened. Ex-type strains are in bold, and the newly introduced taxa are in red.

**Table 2 jof-08-00076-t002:** Synopsis of morphological characteristics, habitats and hosts compared across *Melomastia* species.

Taxa	Asci Size (μm)	Ascospores		Septa in Ascospores	Sheath	Habitats & Host Records	Location	References
		Morphology	Size (μm)					
*Melomastia aquatica*	185–230 × 7–9	Fusiform	26–34 × 6–8	3	Yes	Freshwater/Unknown	China	[9]
*M. chromolaenae*	135–160 × 7–8	Fusiform	29–35 × 4.5–6	9–11	No	Terrestrial/*Chromolaena odorata*	Thailand	[42]
*M. clematidis*	115–160 × 4–7	Broad fusiform with acute ends	13–20 × 3.8–5	3	Yes	Terrestrial/*Clematis sikkimensis*	Thailand	[19]
*M. distoseptata*	126.7–146.2 × 4.7–6.3	Fusoid, obtuse ends	19.7–24.9 × 4.3–5	3	No	Terrestrial/Unknown	Andaman	[15]
*M. fulvicomae*	70–90 × 4–6	Broad fusiform with rounded ends, ends acute	9–15 × 3.5–5.5	2–3	Yes	Terrestrial/*Clematis fulvicoma*	Thailand	[19]
*M. fusispora*	200–231 × 7.6–9.2	Fusiform	27.5–32 × 6.5–7.5	3	Yes	Terrestrial/*Olea europaea*	China	this study
*M. italica*	120–190 × 5.1–8.9	Ellipsoidal	8.8–10.5 × 2.8–4.11	2	Yes	Terrestrial/*Vitis vinifera*	Italy	[18]
*M. mangrovei*	154–216 × 8.5–14	Fusiform	26–33 × 6–8	7–9	Yes	Intertidal/*Rhizophora* sp.	Thailand	[21]
*M. maolanensis*	(95–) 103–118 (–122) × 4–5.5	Fusiform with round ends	13.5–18 × 3.5–4.5	3	No	Terrestrial/Unknown	China	[28]
*M. marinospora*	190–240 × 10–12	Cylindrical with acute poles	25–31 × 7.5–10	3	Yes	Intertidal/*Kandelia candel*	Brunei	[21]
*M. neothailandica*	165–190 × 10–12	Ellipsoidal	26–28 × 7.2–8	5	Yes	Marine/*Rhizophora* sp.	Thailand	[43]
*M. oleae*	209–237 × 7.5–9	Fusiform, obtuse ends	28–34 × 6–7	3	No	Terrestrial/*Olea europaea*	China	this study
*M. phetchaburiensis*	190–300 × 8–12	Ellipsoidal	35–40 × 5– 10	1–10	No	Marine/*Rhizophora apiculata*	Thailand	[44]
*M. rhizophorae*	135–160 × 8–10	Ellipsoidal	19–26 × 6– 8	4–6	Yes	Intertidal/*Rhizophora apiculata*	Thailand	[9]
*M. sichuanensis*	101–112.5 × 6.5–7.6	Broad fusiform with rounded ends	15–17.5 × 4.7–5.1	3	Yes	Terrestrial/*Olea europaea*	China	this study
*M. sinensis*	160–220 × 8–10	Cylindrical	18–30 × 5– 8	6–7	No	Terrestrial/*Camellia sinensis*	Thailand	[45]
*M. thailandica*	146– 158 × 7–9	Ellipsoidal	24–32 × 6–8	3–5	Yes	Marine/*Avicennia marina*	Thailand	[46]
*M. thamplaensis*	114–160 × 6–8.5	Fusiform with acute angular ends	19.5–23.5 × 5–6.5	3	No	Terrestrial/Unknown	China	[28]
*M. tiomanensis*	316–333 × 12–17	Spindle-shaped	69–82 × 9–11	6–7	No	Terrestrial/*Rhizophora* sp.	Malaysia	[20]
*M. winteri*	165–189 × 7–8.5	Fusiform with acute ends	25–30 × 5–6.5	3	No	Terrestrial/*Olea europaea*	China	this study

## Data Availability

All sequence data are available in NCBI GenBank following the accession numbers in the manuscript.

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
