# Peer review of "Reassessment of Dyfrolomyces and Four New Species of Melomastia from Olive (Olea europaea) in Sichuan Province, China"

_jof, 2022, doi:10.3390/jof8010076_

Round 1
Reviewer 1 Report
In index Fungorum ... Melomastia is Sordariomycetes while Difrolomyeces is Dothideomycetes ... then ... Morphological terms for Sordariomycetes should be used in the descriptions
Species recover well on phylogeny; However, it would have been desirable to include the type M. mastoid.
In the synonymy order chronologically
Species diagnoses are not included
In the disscusiion abound on the impact of introduced species and the global distribution considering that they could be endophytes

Author Response
In index Fungorum ... Melomastia is Sordariomycetes while Dyfrolomyces is Dothideomycetes ... then ... Morphological terms for Sordariomycetes should be used in the descriptions
Currently, the genera Dyfrolomyces, Melomastia and Pleurotrema are accepted in the family Pleurotremataceae (Wijayawardene et al. 2017, Hongsanan et al. 2020), and Pleurotremataceae has been excluded from Sordariomycetes by Maharachchikumbura et al. 2015. Thus, the taxonomic position of Dyfrolomyces, Melomastia and Pleurotrema are in Dothideomycetes. This revision has not been updated in index Fungorum, however MycoBank has been updated and lists its classification as “Fungi, Ascomycota, Pezizomycotina, Dothideomycetes, Dyfrolomycetales, Pleurotremataceae”.
And what is the difference (M. fusiform) with M. oleae?
In M. fusiform’ notes,we only compared the morphological characteristics and molecular differences between Melomastia fusiform and M. winteri which is the most closely related to M. fusiform in our phylogenetic tree,but in M. oleae’ notes, we compared the differences of M. oleae and M. fusiform, M. oleae and M. winteri, M.oleae and M. thialandicas, respectively due to M. oleae formed a distinct clade sister to the clade containing M. fusispora, M. winteri and M. thamplaensi.
(319-322) Melomastia fusispora is easily differentiated from M. oleae by having non-carbonised peridium and relatively lager ascomata (528 × 572 μm vs. 425 × 506.5 μm). Additionally, tef1-α sequence comparison reveals 48 bp differences without gaps across 875 nucleotides (5.48%) between Melomastia fusispora and M. oleae.
It could be inferred that it is endophyte of Olea europea, therefore is it an introduced species? / These species could be endophytes on Olea europea? abound on the subject
This is an interesting inference! We read the literature about the relationship between endophytes and saprophytes. The Foraging Ascomycete (FA) hypothesis proposes that saprotrophic fungi can utilize leaves both as dispersal vehicles and as resource havens during times of scarcity (Thomas et al. 2016). Nelson et al (2020) verified this hypothesis and introduced the term Viaphyte (literally, "by way of plant") to refer to fungi that undergo an interim stage as leaf endophytes and, after leaf senescence, colonize other woody substrates via hyphal growth. All of Melomastia spp. nov in this study were isolated from the dead branches of Olea europea. It is difficult to define whether these species are endophytes or saprophytes, but what is certain is that when we isolated it was in its saprophytic phase of life history.
How its affect the Chinese Mycobiota or Chinese ecosystem. And the global distribution of Melomastia?
So far, there are seven Melomastia species reported in China, and distributed in Guizhou, Sichuan Hongkong. The limited number of isolates make it impossible to discuss the effect of this group on the Chinese Mycobiota or Chinese ecosystem. In addition, the main contribution of this paper is to describe the new taxa of Melomastia and make the revision of Dyfrolomyces based on the molecular phylogeny and morphology (as well as the fresh collection). To address the distribution of species of Melomastia, we added a sentence to mention the global distribution of Melomastia species in discussion section. “Species of Melomastia have a wide geographical distribution, eg. Africa, China, Germany, Italy, Japan, Poland and United States of America (California)”
Should suggest checking Olea trees in Europe?
Yes, we added this suggestion in the discussion section.
Olives are cultivated in many regions of the world with Mediterranean climates, such as Australia, Chile, Italy, Peru, South Africa and USA (California and Oregon). This study represents the first discovery of Melomastia on the Olea europaea in China.
Referrence
Hongsanan, S.; Hyde, K.D.; Phookamsak, R.; Wanasinghe, D.N.; McKenzie, E.H.C.; Sarma, V.V.; Boonmee, S.; Lücking, R.; Bhat, D.J.; Liu, N.G.; et al. Refined families of Dothideomycetes: Dothideomycetidae and Pleosporomycetidae. Mycosphere 2020, 11, 1553–2107.
Nelson, A., Vandegrift, R., Carroll, G.C., Roy, B.A. Data from: double lives: transfer of fungal endophytes from leaves to woody substrates. figshare. Dataset 2019.
Maharachchikumbura, S.S.N.; Hyde, K.D.; Jones, E.B.G.; McKenzie, E.H.C.; Huang, S.K.; Abdel-Wahab, M.A.; Daranagama, D.A.; Dayarathne, M.; D’souza, M.J.; Goonasekara, I.D.; et al. Towards a natural classification and backbone tree for Sordariomycetes. Fungal Diversity 2015, 72, 199–301, doi:10.1007/s13225-015-0331-z.
Thomas, D.C., Vandegrift, R., Ludden, A., Carroll, G.C., Roy, B.A. Spatial ecology of the fungal genus Xylaria in a Tropical Cloud Forest. Biotropica 2016, 48, 381–393.
Wijayawardene, N.N.; Hyde, K.D.; Rajeshkumar, K.C.; Hawksworth, D.L.; Madrid, H.; Kirk, P.M.; Braun, U.; Singh, R.V.; Crous, P.W.; Kukwa, M.; et al. Notes for genera: Ascomycota. Fungal Diversity 2017, 86, 1–594, doi:10.1007/s13225-017-0386-0.
Reviewer 2 Report
The paper is devoted to the taxonomic evaluation of two genera from the family Pleurotremataceae and the description of four new species collected from the olive trees in China. The taxonomic work includes thorough molecular data based on sequencing the nuclear ribosomal and protein-coding genes of the revised group of species, as well as their detailed morphological descriptions supplied with excellent pictures.
On my mind, there are only few aspects, which should be clarified.
In Results (lines 232-235), it is written that in spite of the fact that molecular data had indicated Melomastia and Dyfrolomyces as two distinct genera, the authors preferred to base on the morphological similarity of these genera and to combine them in a single genus. I think such point of view is acceptable. However, in Discussion the authors wrote that considering both the morphological comparison and multi-gene phylogenetic analysis, Melomastia and Dyfrolomyces are congeneric, and similarly in Conclusion, it is written that Dyfrolomyces was synonymized under Melomastia based on molecular phylogeny and morphology. The first and the two last statements look rather contradicting in the relation to molecular phylogeny. Therefore, the authors should clarify this aspect more clearly.
Table 2. The meaning of "Y" and "N" should be determined, or just written "yes" and "no".
I suggest placing the habitat description like "Saprobic on dead branches of Olea europaea" after "Culture characteristics"
All other minor suggestions are inserted into the PDF version of manuscript, which is attached.
Author Response
- In Results (lines 232-235), it is written that in spite of the fact that molecular data had indicated Melomastia and Dyfrolomyces as two distinct genera, the authors preferred to base on the morphological similarity of these genera and to combine them in a single genus. I think such point of view is acceptable. However, in Discussion the authors wrote that considering both the morphological comparison and multi-gene phylogenetic analysis, Melomastia and Dyfrolomyces are congeneric, and similarly in Conclusion, it is written that Dyfrolomyces was synonymized under Melomastia based on molecular phylogeny and morphology. The first and the two last statements look rather contradicting in the relation to molecular phylogeny. Therefore, the authors should clarify this aspect more clearly.
Based on this phylogenetic tree, we have 2 taxonomic schemes, the first one: identified Clade A as Melomastia, Clade B as Dyfrolomyces, the second: identified Clade A and Clade B as one genus, combined with the morphological features, we selected the second scheme. This is our originally idea, but there are some ambiguitys in our representation. We have rewritten this part
However, the taxa in these two clades have no noticeable morphological differences. We conclude the Melomastia and “Dyfrolomyces” are congenerics based on molecular phylogeny and morphology. Further studies with fresh collections are needed to resolve the taxonomic relationships in Pleurotremataceae and its sexual-asexual connections.
- I suggest placing the habitat description like "Saprobic on dead branches of Olea europaea" after "Culture characteristics"
Following the general format of “Taxonomy”, we would prefer to assign the habitat information in the description section. Hence, we did not provide it after the Culture characteristics